# Folate Repletion after Deficiency Induces Irreversible Genomic and Transcriptional Changes in Human Papillomavirus Type 16 (HPV16)-Immortalized Human Keratinocytes

**DOI:** 10.3390/ijms20051100

**Published:** 2019-03-04

**Authors:** Claudia Savini, Ruwen Yang, Larisa Savelyeva, Elke Göckel-Krzikalla, Agnes Hotz-Wagenblatt, Frank Westermann, Frank Rösl

**Affiliations:** 1Division of Viral Transformation Mechanisms, German Cancer Research Center (DKFZ), 69120 Heidelberg, Germany; claudia.savini2@gmail.com (C.S.); ruwen.yang@dkfz-heidelberg.de (R.Y.); e.goeckel-krzikalla@dkfz-heidelberg.de (E.G.-K.); 2Division of Neuroblastoma Genomics, German Cancer Research Center (DKFZ), 69120 Heidelberg, Germany; l.savelyeva@kitz-heidelberg.de (L.S.); frank.westermann@dkfz-heidelberg.de (F.W.); 3Omics IT and Data Management, German Cancer Research Center (DKFZ), 69120 Heidelberg, Germany; hotz-wagenblatt@dkfz-heidelberg.de

**Keywords:** human papillomavirus, anogenital cancer, karyotype, genomics, transcriptomics, micronutrients

## Abstract

Supplementation of micronutrients like folate is a double-edged sword in terms of their ambivalent role in cell metabolism. Although several epidemiological studies support a protective role of folate in carcinogenesis, there are also data arguing for an opposite effect. To address this issue in the context of human papillomavirus (HPV)-induced transformation, the molecular events of different folate availability on human keratinocytes immortalized by HPV16 E6 and E7 oncoproteins were examined. Several sublines were established: Control (4.5 µM folate), folate deficient (0.002 µM folate), and repleted cells (4.5 µM folate). Cells were analyzed in terms of oncogene expression, DNA damage and repair, karyotype changes, whole-genome sequencing, and transcriptomics. Here we show that folate depletion irreversibly induces DNA damage, impairment of DNA repair fidelity, and unique chromosomal alterations. Repleted cells additionally underwent growth advantage and enhanced clonogenicity, while the above mentioned impaired molecular properties became even more pronounced. Overall, it appears that a period of folate deficiency followed by repletion can shape immortalized cells toward an anomalous phenotype, thereby potentially contributing to carcinogenesis. These observations should elicit questions and inquiries for broader additional studies regarding folate fortification programs, especially in developing countries with micronutrient deficiencies and high HPV prevalence.

## 1. Introduction

Folates are a group of B vitamins that provide one-carbon availability for DNA methylation and nucleotide biosynthesis [1]. Micronutrient supplementation and metabolic pathways are highly interconnected and can therefore substantially determine the cellular phenotype [2]. Considering a holistic concept of multi-step carcinogenesis, several studies underline the impact of lifestyle, such as diet on cancer risk and final tumor development [3]. Nutrition therefore still represents a not well understood and unpredictable factor that, together with genetic alterations and/or infections with potential tumor viruses, could affect the individual cancer susceptibility [4]. Nowadays acute nutrient deficiency is mostly restricted to developing countries where there is also a high prevalence of human papillomavirus (HPV) infections and anogenital cancer [5]. In fact, as shown by several epidemiological studies, low folate levels are considered to be an additional risk factor in cervical cancer development [6].

This is of clinical relevance since cervical cancer is one the most frequent malignancies in women worldwide and “high-risk” HPV types such as 16 and 18 are the etiological agents [7]. Viral transforming potential is conferred to the expression of the *E6* and *E7* oncogenes [8] that can attack central hubs within a cellular network to get a selective growth advantage [9].

Meta-analyses of a large number of randomized control trials about the role of folate fortification and cancer incidence provided numerous data showing a range from either no [10], protecting [11], or cancer-promoting effects [12]. Conversely, in the case of folate deficiency, there are also many studies showing a correlation toward higher cancer risk [13].

This indicates that the impact of folate modulation is multifactorial, depending on the type and stage of cancer [14], the folate dosage [15], the presence of additional risk factors [16], or nucleotide polymorphism within the methylenetetrahydrofolate reductase (*MTHFR*) and thymidylate synthase (*TS*) genes [17], respectively. In fact, folate deficiency can inhibit the growth of already existing pre-neoplastic lesions, whereas its supplementation may have a promoting effect [10]. These results raise some concerns about overall implementation programs of folic food fortification [3], especially when supplementation should be applied after preceding deficiency. Inspecting the literature, however, there are only a few studies that explicitly addressed this question [18].

The present report is an attempt to fill this gap by studying the role of folate in the context of HPV-linked carcinogenesis in molecular terms. As a proof-of-concept experiment, we analyzed the phenotypic changes and the molecular events of folate modulation in vitro on human keratinocytes, immortalized by HPV16 *E6* and *E7* oncogene expression. We show that supplementation to initial levels cannot compensate preceding folate-deficient effects. Considering that cancer is a multi-step process, increased cellular proliferation, impaired DNA repair fidelity, clonogenicity, and selection of unique chromosomal aberrations may potentially drive immortalized cells towards transformation. Hence, folate fortification programs to complement micronutritional deficiencies should be surveyed by broad prospective epidemiological and molecular studies, especially in developing countries with high HPV prevalence.

## 2. Results

### 2.1. Folate Deficiency and Repletion Generate Phenotypes with Altered Metabolism and Proliferation

To investigate the effects of different folate availability, human keratinocytes immortalized by HPV16 *E6* and *E7* (HFK16E6E7) were used as model system [19]. As shown in the schematic overview (Figure 1A), three sublines were established: (a) The original HFK16E6E7 cell line, grown in medium with standard folate content (referred as FC); (b) cells adapted to growing in low folate levels (referred as FD) for 15 weeks to ensure a stable in vitro phenotype [20]; and (c) FD cells reconstituted with standard folate medium (referred as FR). To confirm the impact of folate modulation on cell metabolism, total homocysteine levels were measured. This metabolite is a common marker that inversely correlates with folate levels [21]. As shown in Figure 1B, FC cells exhibited low and stable total homocysteine levels, while FD cells revealed an increase of homocysteine of more than ten-fold. Reversibility of homocysteine levels in FR cells could be discerned after 9 weeks of folate repletion (week 24, Figure 1B), being in line with another study also showing a functional link between homocysteine levels and folate availability [22].

Using cell viability assays, FR cells displayed a significant increase in proliferation rate as compared to FD and FC cells, while the difference between FD and FC cells was only marginal (Figure 1C). In HPV-positive cells, *E6* and *E7* oncogene expression is decisive to maintain a proliferative phenotype [23]. To examine whether the higher growth rate was related to an enhanced oncogene expression, levels of E6, E7, and their known targets p53 and pRb were investigated [7]. As shown in Figure 2, there are only minor variations in E6 and E7 oncoprotein expression. p53 and pRb as major downstream targets for proteasomal degradation remained unchanged in FC, FD, and FR cells, therefore not accounting for different cellular growth behaviors.

To further characterize features of cells under different folate culture conditions, colony formation assays were performed [24]. As depicted in Figure 3A–D, FD cells formed only a few colonies while FR cells exhibited a significant increase in number and size in comparison with FC cells. These results indicate that folate repletion can lead to a selection of cells with higher clonogenicity [25].

### 2.2. Folate Deficiency Results in Irreversible DNA Damage, Nucleotide Loss, and Misincorporation

Folate deficiency not only disturbs the whole cell metabolism but also exerts cellular stress that can affect chromosomal stability and the DNA damage response [26]. Therefore, the status of DNA damage and, in particular, its potential reversibility under different folate conditions was investigated. For this purpose, we examined γH2AX foci formation, a key marker of double stranded DNA breaks [27]. As shown in Figure 4, there was a substantial increase in γH2AX foci formation in FD cells when compared to the original FC cell line. This effect is linked to uracil misincorporation into DNA due to a reduced nucleotide pool caused by folate deficiency [26]. Formation of γH2AX foci in FR cells remained at a similar proportion as in FD cells, but was significantly higher than in FC cells. This indicates that, despite the fact that the nucleotide precursor pool was restored upon folate re-supplementation, irreversible DNA damage remained.

Next, in vivo DNA ligation assays were performed, allowing the measurement of the ability to repair double-stranded DNA ends by non-homologous end joining (NHEJ), an important pathway to maintain chromosomal integrity [28]. For this purpose, either blunt-end or 3’-incompatible overhang-end linearized plasmid was transfected into FC, FD, or FR cells to assess their ability to restore circular DNA. Single clones of recircularized plasmids were sequenced and analyzed (Figure 5A). As shown in Figure 5B, the mean of nucleotide loss and misincorporation increases in FD and FR cells, whereby the NHEJ repair showed the most pronounced difference (Figure 5B and Appendix A). These results provide evidence of a long-lasting impairment of DNA repair fidelity in the folate deficient cells that is still maintained in folate repleted cells.

### 2.3. Folate Deficiency Cells Harbor Chromosomal Rearrangements and Folate Repletion Results in a Clonal Selection

To examine whether folate deficiency affects chromosomal stability, karyotype analysis was performed using multicolor FISH (M-FISH) [29]. FC cells had a modal chromosome number of 47 with trisomy of chromosome 20. In all 46 metaphases analyzed, two derivative chromosomes were observed: der(14)t(7;14), a derivative chromosome 14 with a translocated chromosome 7 segment and der(7;10), a derivative chromosome resulting from a whole arm translocation of 7p to 10q (Figure 6A). 26% of metaphases harbored unique translocations or deletions visible on a single metaphase and 9% displayed unrepaired chromosome or chromatid breaks (Figure 6D).

M-FISH karyotyping of FD cells revealed that 64% retained the original karyotype observed in FC cells, whereas 36% of metaphases presented new clonal cell populations with different derivative chromosomes (Figure 6B). Besides changes in clonal composition, 29 additional chromosomal alterations were found in 59% of FD cells, including translocations, deletions, and dicentric chromosomes. 41% of metaphases exhibited acentric chromosome fragments or chromatid breaks (Figure 6D). A comparison of FD with FC cytogenetic profiles revealed a folate deficiency-related increase in chromosomal damage and accumulation of numerous gross chromosomal rearrangements.

To investigate possible changes after repletion, karyotype of FR cells was also assessed by M-FISH. Among 41 metaphases, only 24% displayed the original FC karyotype. The remaining 76% of metaphases arose from a clonal cell population harboring two novel derivative chromosomes: del(10) and der(14)t(7;8;14), a product from a translocation of an extra copy of the distal part of 8q to the der(14)t(7;8) observed in FC cells (Figure 6C). The same karyotype pattern had already been observed in 18% of FD metaphases and thus might provide a growth advantage of this cell population upon folate repletion. Of all metaphases, 10% harbored unique gross chromosomal rearrangements and a single chromatid break was seen in only 2% of cells (Figure 6D). The substantial decrease in novel chromosomal rearrangements and reduction of chromosome breakage in FR cells could be due to a selective growth advantage of a single clone yielding a more homogeneous cell population.

### 2.4. Whole Genome Sequencing Reveals Distinct Genomic Alterations in FC Cells

To survey structural alterations at the sequence level after folate modulation, whole genome sequencing (WGS) of DNA from FC and a subclone of FR cells was performed. 99.97% of mapping ratio was reached at median coverage of approximately 38× (data available at ENA accession number: PRJEB25728). Upon further filtering, SOPHIA analysis showed a higher number of structural variations (SVs) in FC compared with FR cells (Figure 7A). In both FC and FR cells, a cluster of rearrangements on chromosome 14 corresponds to the translocation junction region in der(14)t(7;14) and der(14)t(7;8,14), highlighting the complexity of recombination events at the distal 14q. Figure 7B shows that in total there were 1453 SVs detected in FC and 469 in FR cells. 1453 SVs appear exclusively in FC while 377 SVs appear only in FR cells. FC and FR cells have 92 SVs in common.

In total, 71845 single nucleotide variants (SNVs) exist in FR or FC cells. In particular, FC cells harbor 66736 SNVs, among which 34600 are not present in FR cells, whereas FR subcloned cells have 37245 SNVs, among which 5109 are not found in FC cells (Figure 7C). Figure 7D shows the comparison of minor allele frequency (MAF) for all the SNVs that appeared in FC or FR cells. SNVs that are only present in FC cells have a MAF value equal to 0 in the FR cells and vice versa. The dot enrichment on the x-axis from 0 to 0.25 is due to a large number of SNVs with low MAF in FC cells that are absent in the FR cells, or have an allele frequency below the detection level.

The absence of approximately half of FC-derived SNVs in FR cells, as well as the reduced number of SVs in FR compared to FC cells, can be explained by natural genomic heterogeneity of the non-clonal FC cell population versus the subclonal FR population propagated from a single cell. However, the presence of SNVs that are exclusively seen in FR cells might be mainly caused by increased DNA damage and improper repair under folate deficiency.

WGS data of FR cells refines M-FISH analysis, demonstrating a copy gain of the distal part of the long arm of chromosome 8 starting from chr8:116439758 to the telomere. This gain results from an 8q translocation to the telomeric portion of the 7p material (chr7:506861) of der(14)t(7;8) derivative chromosome presented in FC cells. As shown in Figure 8B, coverage based on allele frequency of SNVs increased starting from the 116 Mb position on chromosome 8 in FR cells that are not present in the FC population (Figure 8A). Comparison with RNA-Seq data showed increased transcription activity in FR vs FC cells on chromosome 8, in accordance with its copy gain upon folate repletion (Figure 8C).

Overall, WGS data analysis showed that restoration of the folate level could result in a cell clone with a novel genomic content associated with the accumulation of multiple DNA sequence alterations that occurred during folate deficiency.

### 2.5. DNA Alterations in Folate Repleted Cells Can Disrupt Gene Structure

To evaluate the frequency of gene structure alterations observed exclusively in FR cells, 120 high-score events encompassing protein coding genes were extracted from SOPHIA results (Appendix A). Consistent with the induction of breakage at fragile sites (FS) upon folate deficiency [31], 24% of these alterations affected common FS, including the most active common FS genes *FHIT*, *PARK2*, and *WWOX* in the human genome.

The majority of the genetic rearrangements were restricted to intronic sequences, however, 33.3% resulted in partial or whole exon deletions (shown in bold in Appendix A). Considering the acquired growth advantage of FR versus FC cells, we examined genes with oncogenic potential for which structural damage could underlie their aberrant expression and function. At least 12 genes (marked in grey in Appendix A) reported to be involved in carcinogenesis showed irreversible alterations within the coding gene sequences. Exonic deletions were detected in the *PPP2R5A*, *FHIT*, *LRP1B,* and *OPCML* tumor suppressor genes, as well as in genes implicated in cellular proliferation (*CCNB1*, *ANAPC5*, *PIK3C2A*) and DNA repair (*SMARCA5*). Besides the intragenic deletions, a large 204 kbp duplication on chromosome 5 (chr5:65111984—65316190) resulted in the formation of the *NLN–ERBB2IP* fusion gene, which has been associated with breast invasive carcinoma [32]. Moreover, the translocation of 8q to der(14)t(7;8) observed in FR cells disrupts the *TRPS1* gene, a transcriptional regulator involved in the control of cell cycle progression [33].

Although we cannot conclude whether these genetic alterations are responsible for the proliferation advantage of the FR cells, our WGS data demonstrate the occurrence of irreversible changes in a number of cancer genes that can be critical for malignant progression.

### 2.6. Folate Deficiency and Repletion Show Gene Expression Variations

To identify gene expression differences among the cells, RNA-Seq analysis was performed. Following bioinformatic analysis of primary processing of the RNA-Seq data, a comparative analysis among genes in FC, FD, and FR cells was carried out. Genes with at least 2-fold expression variation and with a *q*-value lower than 0.05 were selected. To compare the transcriptomes under different folate conditions, a Venn diagram shows the number of differentially expressed genes (Figure 9). The majority of variation was observed in FD vs. FC and FR vs. FD groups. In particular, increased expression occurs upon folate deficiency (FD vs. FC), whereas decreased expression is predominant upon repletion (FR vs. FD). This indicates that with the generation of the FR cell line, a high number of genes with varying expression in the FD cell line were reverted to similar levels as in the original FC cells.

A subset of genes that show varying expression in FR in comparison to FC cells were further annotated based on their known functions and their relevance to cancer development and potential candidates were selected (Table 1) for validation by quantitative real-time PCR (qPCR) (Figure 10). Consistent with the RNA-Seq results, the qPCR data show significant differential expression of the selected genes. In particular, the validated genes are also representative of four main profiles of expression that could be categorized as follows (Figure 10A–D): Firstly, genes with decreased expression exclusively in the FR cell line are *CDKN1C*, *H2AFY2*, *LEF1*, and *TBX18* (Figure 10A). Secondly, genes where decreased expression was maintained after depletion and not significantly increased in cells with re-supplemented folate: *DIRAS3* and *USP44* (Figure 10B). Thirdly, genes with an opposite expression pattern in folate deficiency and repletion cell lines: *SORBS2*, *DCN*, *SLFN11*, *RUNX3*, *GALC*, and *HSPB7* (Figure 10C). Notably, decreased gene expression in FR cells has been previously described to play a tumor suppressive role (summarized in Table 1 and references therein). Lastly, enhanced expression in FD and FR cell lines was found in the following genes: *FAP*, *KLK5*, *KLK10*, and *PIM1* (Figure 10D). In contrast to the latter (Figure 10C), these genes have putative or already known oncogenic functions and their expression was increased upon folate deficiency and even further augmented upon repletion.

In summary, these results show that altered availability of folate can influence cellular stability and transcription that can affect, depending on the selection pressure in a particular environment, different cellular pathways and functions. This may be supportive for future studies to elucidate the molecular changes acquired after undergoing a folate deficiency period.

## 3. Discussion

As shown by meta-analyses of randomized control trials, there is still controversy about the role of folate fortification and cancer incidence, ranging from no [10], protecting [11], to cancer-promoting effects [12]. This reflects the enormous complexity of cancer and the different micro-evolutionary processes that are either predisposing or preventing the outgrowth of a tumor [34]. Although holistic thinking is necessary to understand human carcinogenesis, reductionist methods are indispensable as a bottom-up approach to study the impact of folate, known to be involved in many metabolic pathways [3]. Moreover, an open topic that is still not sufficiently addressed in broader studies is the impact on cells when folate is repleted after a period of deficiency.

Addressing this question, we showed that long-term adaptation of HPV16 immortalized human keratinocytes [19] in folate deficient medium and subsequent repletion led to considerable changes in their phenotypes. Although there was no interference with the expression of early viral genes, it is noteworthy that folate deficiency can already perturb normal virus maturation after an initial HPV infection [35]. As also shown in our cell system, folate depletion, for instance, increases the intracellular homocysteine level, known to trigger homocysteinylation of the heterogeneous nuclear ribonucleoprotein E1 (hnRNP E1), an RNA-binding protein involved in RNA processing [20]. hnRNP E1 in turn negatively impairs the expression of the papillomavirus L2 capsid protein so that replicated viral DNA cannot be encapsidated. Interference with the natural permissive cycle in any form can favor integration [35], leading to dysregulated viral oncogene expression [36]. Hence, as already shown by epidemiological studies [37], such a scenario could therefore represent an early event in HPV-induced carcinogenesis by forcing viral integration during the course of a natural infection [38].

Moreover, removal of folate is also known to alter DNA stability by inducing replication stress and DNA damage as consequences of a nucleotide pool imbalance and uracil misincorporation, independently of any viral infection [39]. Excision of uracil from DNA can create DNA breaks that are constantly and increasingly present in case of a continuous limited nucleotide pool and uracil misincorporation [40]. The significant increase in γH2AX foci formation (Figure 4), a known marker of dsDNA breaks [41] is therefore in line with previous studies [39]. This could be an additional step on the way to cervical cancer, namely the temporal combination of folate deficiency in conjunction with persistent HPV, in fact accounting for a higher cancer incidence [37].

Remarkably, in cells re-adapted to standard folate levels, γH2AX foci formation could not be reversed and was even increased (Figure 4). Since γH2AX foci should disappear shortly after DNA repair [42], their persistence after repletion despite nucleotide pool reconstitution could indicate that cells are permanently under genotoxic stress due to perturbed DNA damage repair pathways during folate depletion and ongoing viral oncogene expression [26].

To test the fidelity of DNA damage repair under these conditions, in vivo ligation assays were performed. This allowed monitoring double strand DNA repair of either sticky- or blunt-end plasmids by non-homologous and microhomology-mediated end-joining pathways [43]. As shown in Figure 5, not only folate deficiency but also repletion showed higher nucleotide loss and misincorporation as compared with control cells. This suggests that the DNA damage repair machinery is continuously compromised during and after folate deficiency, where the inability to correctly repair incompatible 3′-ends was even more affected. The reason for persisting DNA damage is still unclear and could also be related to chromatin remodeling that limits or delays the accessibility of the DNA repair machinery [44]. Indeed, performing quantitative chromatin conformation measurements by super-resolution fluorescence microscopy already showed that folate deficiency leads to the chromatin relaxation as detected by decreased histone H3K9me3 heterochromatin after immunostaining in 3D-conserved cell nuclei. In repleted cells, a relative relaxation of the chromatin to an intermediate situation between FC and FD cells was still maintained, indicative of the continuous accumulation of DNA damage and attempts of repair. Hence, folate modulation can not only delay the recruitment of repair proteins but also affect the nucleosome organization of the DNA [45].

Since cancer is an evolutionary process [34], changes in the micronutritional milieu always result in a specific selection pressure on the respective cells, enabling only the fittest to adapt and to proliferate [46]. Karyotype analysis shows that cells adapted to folate deficiency harbor chromosomal heterogeneity (Figure 6). Although it is claimed that the expression of HPV16 E7 may lead to centrosome overduplication and chromosomal instability [47], FC cells (Figure 6A) only showed minor rearrangements, harboring two marker chromosomes as indicated. This is in line with a recent study, demonstrating that HPV types with high immortalization capacity (e.g., HPV16 and HPV18) and physiological levels of oncogene expression have low chromosomal instability in immortalized keratinocytes [48]. However, cells under folate deficiency and constant impairment of the DNA damage response had more karyotype changes, indicative of the accumulation of new clones within the cell population harboring a novel genomic content (Figure 6B). Remarkably, repletion of folate led to the clonal selection of cells harboring derivative chromosomes already existing under folate deficient conditions while the number of gross chromosomal rearrangements was reduced (Figure 6C). The underlying mechanisms on how initial folate depletion is exactly affecting karyotype stability are still poorly understood. However, as shown in a recent study, chromosome segregation during mitosis, controlled by the spindle assembly checkpoint pathway, seems to be regulated by the folate level and may therefore account for chromosomal instability [49]. Notably, copy gain of chromosome 8 and its translocation (Figure 8B) is also found in other malignancies such as leukemia, breast, and ovarian cancers [50,51,52]. In cervical squamous cell carcinoma, a copy number increase was also mapped to chromosome 8q [53], to a region in proximity of the c-Myc locus, known to be a preferential site of HPV integration [54]. Moreover, as shown by transcriptome analysis, chromosome 8 copy gain also correlates with increased transcripts in folate repleted cells (Figure 8C) which may either directly or indirectly account for enhanced proliferation and clonogenicity (Figure 1 and Figure 3) and was absent in control and folate deficiency cells (Appendix A).

Chromosomal rearrangements were further confirmed by whole genome sequencing of FR cells with del(10) and der(14)t(7;8;14) when compared to the original control population (FC). FD cells were excluded due to increased heterogeneity as already shown by quantifying the M-FISH data (Figure 6D). Different folate availability did not only cause chromosomal instability, but also affected gene expression. Transcriptome profiling revealed a broad spectrum of deregulated genes (Table 1), whose expression may have either oncogenic or tumor suppressive functions. This suggests that changes in folate levels induce alterations across the transcriptome, potentially promoting a harmful phenotype in terms of sustained impairment of the DNA repair machinery (Figure 4 and Figure 5) and subsequent karyotypic alterations (Figure 6). Considering the evolutionary concept of carcinogenesis [34], our results underline not only the high plasticity of a cellular network to compensate environmental changes [55], but also show an irreversible phenotype when the original external milieu was reconstituted. This is mainly due to the accumulation of a particular karyotype [46], which in turn regulates the expression of pro-and anti-proliferative genes [56] (Table 1, Figure 10). In other words, there is certainly an inherent causal and chronological order of pathway (de)activation within the respective cell system during folate modulation, but the outcome is contingent in terms of a stochastic chromosomal landscape formation in different cells and their selective advantage [57]. This concept is currently hyped in the context of personalized medicine to understand divergent evolutionary trajectories [58], both in patient-derived xenografts and in cancer patients in general [59].

Since there is still limited knowledge about the effect of folate deficiency and its reconstitution on cells, this study may provoke questions about general folate fortification after a period of deficiency. Although folate is an important micronutrient that is highly interconnected with central regulatory pathways [2], its deficiency can force HPV integration after early infection [35,38]. However, reconstitution may enhance other oncogenic events, especially in cells already persistently infected with HPV. Anti-folate drugs in cancer treatment may create a similar situation, providing a basis for clonal selection. Such a scenario may explain an increased risk of secondary malignancy after therapy with methotrexate [26].

Concerns related to a dual folate effect on different stages of cell transformation still remain in many countries where a mandatory folate fortification was initiated to prevent neuronal tube defects [60]. Despite the WHO-stated absence of a reliable association of folate with a negative health impact [61], there is a growing number of studies reporting different side-effects across populations, ages, genders, and genetic backgrounds [62]. The potential impact on cancer promotion needs to be considered in the long-term after folate fortification and the molecular changes induced by folate modulation still require further investigations.

## 4. Materials and Methods

### 4.1. Cell Culture

Human foreskin keratinocytes were retrovirally transduced expressing the E6 and E7 open-reading frames of HPV16 (HFK16E6E7) under the control of Moloney murine leukemia virus (MoMuLV) LTR [19] and grown on NIH3T3 cells as feeder [63]. HFK16E6E7 cells (referred to as folate control (FC) cells) were used to generate folate deficiency cells (FD) that were adapted to grow in folate deficient medium for 15 weeks. Folate repletion cells (FR) were generated by culturing FD cells in standard culture medium for 9 weeks. FC and FR cells were grown in FAD (Flavin-Adenin-Dinukleotid) medium containing 3 parts Ham’s F-12, 1 part Dulbecco modified Eagle medium (DMEM), 2.5% fetal bovine serum (FBS), insulin (5 μg/mL), epidermal growth factor (10 ng/mL), cholera toxin (8.4 ng/mL), adenine (24 μg/mL), and hydrocortisone (0.4 μg/mL). This medium contains 4.5 µM of folate. The medium used for FD cells has the same composition as the medium used for FC and FR cells, except for Ham’s F-12 and DMEM that are folate-free versions: [Folate-free Ham’s F-12 (ref. P04-15549 from PAN-Biotech, Aidenbach, Germany), folate-free DMEM (ref. AL007F from HIMEDIA), 2.5% fetal bovine serum (FBS), insulin (5 μg/mL), epidermal growth factor (10 ng/mL), cholera toxin (8.4 ng/mL), adenine (24 μg/mL), and hydrocortisone (0.4 μg/mL)]. This medium contains 0.002 µM of folate contributed from 2.5% FBS.

### 4.2. Homocysteine Level Determination

Media supernatants were collected after 3 days and homocysteine levels were measured at the Metabolomics Core Technology Platform, Heidelberg by the HPLC/UPLC system, as previously described [64].

### 4.3. MTT Assay

1 × 10^4^ cells/well were seeded in 96 well plates and incubated for 24, 48, and 72 h. MTT (3-(4,5-dimethylthiazol-2-yl)-2,5-diphenyl-tetrazolium bromide, Sigma, Munich, Germany) was added to a concentration of 5 mg/mL for an additional 3 h. The supernatant was discarded and 100 μL isopropanol was added and incubated for 15 min at room temperature on a shaker. The absorbance was measured at 570–690 nm using SPECTROstar Nano microplate reader (BMG Labtech, Ortenberg, Germany).

### 4.4. Colony Formation Assay

500 cells/9.5 cm^2^ were cultured for 14 days. Colonies were fixed with 10% glutaraldehyde in PBS (Phosphate-buffered saline) and stained with crystal violet (0.05% solution). The dye was solubilized in 33% acetic acid and the optical density was measured at 570 nm using SPECTROstar Nano microplate reader (BMG LABTECH). Images of cells forming colonies were taken with 10× objective EVOS XL Core Cell Imaging System (Thermo Fisher Scientific, Waltham, MA, USA) and the number of colonies was counted.

### 4.5. Immunofluorescence

FC, FD, and FR cells were grown on cover slides and fixed with 4% paraformaldehyde followed by permeabilization with 0.5% Triton X-100 at room temperature. Blocking was performed in a solution of 2% BSA (Bovine Serum Albumin) in PBS, 0.1% Triton X-100, and 0.3 M glycin. Staining was carried out at room temperature with anti-γH2AX-phosphorylated Ser139 antibody (Millipore, Darmstadt, Germany, diluted 1:250). AlexaFluor488 goat anti-mouse was used as a secondary antibody (Invitrogen, diluted 1:450). DNA was stained with Hoechst 33258 in PBS 1:1000 *w*/*v* (Sigma). Mowiol (Sigma) was used as mounting medium. Immunofluorescence staining was visualized by Olympus FluoView™ FV1000 Confocal Microscope (60× magnification). All images were collected and processed using the FluoView Software (Olympus, Hamburg, Germany) and γH2AX foci were counted using ImageJ software (National Institutes of Health, Bethesda, MD, USA; http://imagej.nih.gov/ij/).

### 4.6. In Vivo DNA-Ligation Assay

In Vivo DNA-Ligation Assay was performed as previously described [65]. Briefly, 2 μg of EGFP-N1 plasmid (Clontech) was linearized either by SacI and KpnI or by Ecl136II and SmaI and transfected using jetPRIME (Polyplus) into FC, FD, and FR cells. Recircularized DNA was recovered after 48 h by Hirt extraction [66] and the junctions were PCR-amplified (400 ng of DNA) using primers Fw: 5′-GTAACAACTCCGCCCCATT-3′ and Rv: 5′-GTTTACGTCGCCGTCCAG-3′, following the DreamTaq PCR Master Mix (2×) protocol (Thermo Fisher Scientific). The PCR product was quantified using NanoDrop^®^ ND-1000 and 12 ng was cloned in pGEM^®^-T Easy vector, (pGEM^®^-T Easy Vector System I, Promega, Mannheim, Germany). DH5-Alpha bacteria (50 µL) were transformed using 2 µL of the ligation mixture. The bacteria suspension was transferred to an LB-Agar plate with ampicillin (100 µg/mL) X-Gal/IPTG (0.1 M) and incubated overnight at 37 °C. Colony PCR of positive clones was performed using T7: 5′-TAATACGACTCACTATAGGG-3′ and SP6: 5′-ATTTAGGTGACACTATAG-3′ primers followed by Miniprep preparations and subjected to Sanger sequencing (primers used: Fw: 5′-GTAACAACTCCGCCCCATT-3′ and Rv: 5′-GTTTACGTCGCCGTCCAG-3′).

### 4.7. Karyotype Analysis by Multicolour FISH (M-FISH)

Karyotypes of FC, FD, and FR cells were obtained using the commercially available 24xcyte multicolor FISH probe mix (MetaSystems, Altlussheim, Germany). DNA denaturation and hybridization to metaphase spreads were performed following the manufacturer’s recommendations. Slides were viewed with a Zeiss Axio Imager. Z1 microscope and karyotypes constructed with ISIS FISH imaging software (MetaSystems).

### 4.8. Immunoblotting

Proteins were extracted using a buffer composed of: 10 mM PIPES; 300 mM NaCl; 1 mM EDTA; 300 mM sucrose; 1 mM MgCl_2_; 0.5% TritonX-100; cOmplete™, EDTA-free Protease Inhibitor Cocktail 1x (Roche, Mannheim, Germany); DTT 1 mM; NaF 1 mM; Na_3_VO_4_ 0.2 mM. Protein quantification was done using Protein Assay Dye Reagent Concentrate (Bio-Rad, Munich, Germany). The Western blotting analysis was performed using 40 µg protein and incubated with the following antibodies: HPV16 E6 (2 µg/mL, #849 Arbor Vita Corporation); HPV16 E7 NM2 (kindly provided by Prof. Martin Müller, DKFZ); p53 (DO-1, Santa Cruz-126); pRb (C-15, Santa Cruz-50); β-Actin clone C4 (Millipore 691001). Intensities of bands were quantified by ImageJ.

### 4.9. Real-Time qPCR

Total RNA was extracted using RNeasy Mini Kit (Qiagen, Düsseldorf, Germany) and reverse transcribed using RevertAid Reverse transcriptase (Thermo Fisher Scientific). Quantitative real-time PCR was performed using primers listed in supplementary Table 1. The iTaq Universal SYBR Green Supermix and CFX96 Touch Real-Time PCR Detection System (Bio-Rad) were used.

### 4.10. Whole Genome Sequencing

FR cells were diluted to single cell distribution in 60 mm culture dishes. Colonies originating from single cells were transferred to individual wells by using paper cloning disks and subsequently amplified. Karyotype analysis was performed and the clone harboring del(10) and der(14)t(7;8;14) was used for WGS. Briefly, DNA was extracted from FC cells and from FR single clone-resulting cells harboring del(10) and der(14)t(7;8;14) using AllPrep DNA/RNA Mini kit (Qiagen). Library preparation and whole genome sequencing were performed by the DKFZ Genomics and Proteomics Core Facility. Briefly, DNA libraries were prepared using 100 ng genomic DNA of each sample with the TruSeq^®^ Nano DNA Library Prep (Illumina, San Diego, CA, USA). The sequencing was performed with Illumina HiSeq X platform applying paired-end reads of 150 bp. A coverage of minimum 30× was ensured for each sample.

### 4.11. Bioinformatic Analysis of Whole Genome Sequencing

Paired end sequencing FastQ files were aligned against the human reference genome hg19 including decoy sequences (hs37d5) using the standard PCAWG alignment workflow (https://github.com/ICGC-TCGA-PanCancer/Seqware-BWA-Workflow) [67] utilizing bwa-0.7.8 mem with default parameters, with the exception of the option “-T 0”. SNV calling was performed without using paired germline control samples using the DKFZ somatic SNV calling workflow which is based on samtools mpileup/bcftools (0.1.19), and initial results were filtered by an in-house pipeline to keep the high-confidence mutations in coding regions as previously described [68]. Furthermore, the SNVs in dbSNP with “common = 1” tag or SNVs with frequencies higher than 1% in ExAC 0.3.1 were filtered. Structural variations (SV) were called using SOPHIA algorithm, a workflow as previously described [68]. Briefly, SOPHIA uses supplementary alignments as produced by bwa-mem. These are candidate chimeric alignments originating from “split reads” and indicate a possible underlying SV. SOPHIA uses only high-quality reads that do not fall on poorly mappable regions or consist of low-quality base calls. SOPHIA uses these reads and further filters the results by comparing them to a background control set of sequencing data obtained using normal blood samples from a background population database of 3261 patients from published The Cancer Genome Atlas (TCGA) and International Cancer Genome Consortium (ICGC) studies and both published and unpublished DKFZ studies, sequenced using Illumina HiSeq 2000, 2500 (100 bp) and HiSeq X (151 bp) platforms and aligned uniformly using the same workflow used in this study. An SV is discarded if: It has more than 75% read support from low quality reads; the second breakpoint of the SV is unmappable in the sample and in 10 or more background control samples; an SV with 2 breakpoints has one present in at least 92 control samples (3% of the control samples); both breakpoints have less than 5% read support.

### 4.12. RNA Sequencing

RNA from three independent extractions was used to generate a cDNA library for sequencing with Illumina HiSeq 2000 v4 platform, generating single-reads of 50 bp (DKFZ Genomics and Proteomics Core Facility). Each sample was tagged with a unique barcode prior to making three pooled libraries (FC, FD, and FR cell lines’ samples in each pool) that were sequenced, ensuring approximately 83 million reads per sample. The data provided here have been deposited in NCBI’s Gene Expression Omnibus [69] and are accessible through GEO Series accession number GSE103044.

### 4.13. Bioinformatics Analysis of RNA Sequencing Data

Low quality bases were removed with Fastq_quality_filter from the FASTX Toolkit 0.0.13 (http://hannonlab.cshl.edu/fastx_toolkit/index.html) with 90% of the read with a Phred quality score >20. Homertools 4.7 [70] were used for polyA-tail trimming and reads with a length <17 were removed. PicardTools 1.78 (https://broadinstitute.github.io/picard/) was used to compute the quality metrics with CollectRNASeqMetrics. With STAR 2.3 [71], the filtered reads were mapped against human genome 38 using default parameters. Count data were generated by HTSeq [72] using the gencode.v22.annotation.gtf file (http://www.gencodegenes.org/) filtered for protein coding genes. For the comparison with DESeq2 [73], the input tables containing the replicates for the groups to compare were created by a custom Perl script. For DESeq2, DESeqDataSetFromMatrix was applied, followed by estimateSizeFactors, estimateDispersions, and nbinomWald Test. The results were annotated with gene information (gene symbol, gene type) derived from the gencode.v22.annotation.gtf file. Data were filtered using the following thresholds: Fold change ≥2 and *q*-value <0.05. Venn diagrams were realized by using Bioinformatics and Evolutionary Genomics tools at http://bioinformatics.psb.ugent.be/webtools/Venn/.

### 4.14. Statistical Analysis

The relative error and variation of results for each sample were shown by error bars. The values represent the standard deviation from the mean. Statistical differences and significance were determined using a two-sided Student’s *t*-test. The *p*-values were considered significant if <0.05.

## Figures and Tables

**Figure 1 ijms-20-01100-f001:**
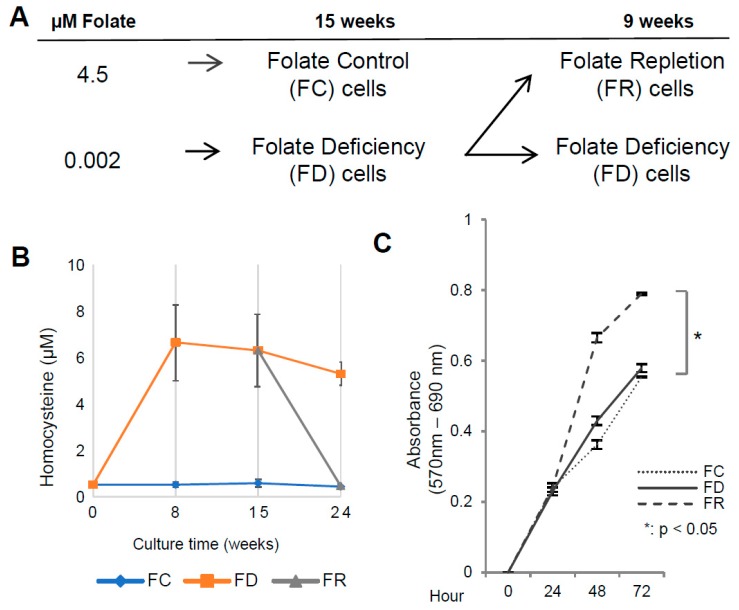
Experimental outline and cellular growth under different folate culture conditions. (**A**) Schematic overview of the establishment of HFK16E6E7 cell lines. Folate control (FC, 4.5 µM), folate deficiency (FD, 0.002 µM), and folate repletion (FR, 4.5 µM) conditions are indicated. (**B**) High-pressure liquid chromatography (HPLC) quantification of total homocysteine levels under different folate conditions as indicated (abscissa; number of weeks). (**C**) MTT assay at week 15 (FC and FD cells) and week 15 + 9 (FR cells). Measurements were carried out at time points 24, 48, and 72 h after seeding. Data shown are mean values of three independent experiments performed in eight replicates * *p* < 0.05; *t*-test HFK16E6E7 FR vs. FC.

**Figure 2 ijms-20-01100-f002:**
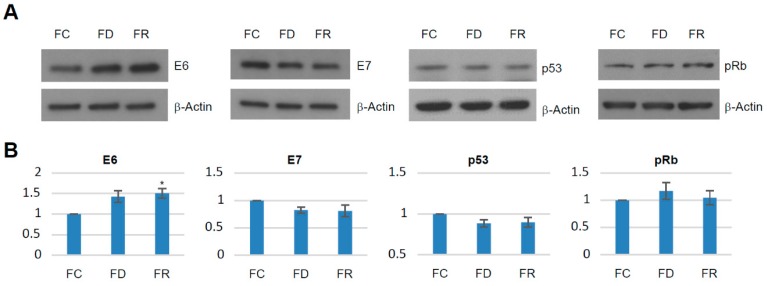
HPV16 E6, E7, p53, and pRb expression levels. 40 µg of total protein extract from each group were used for western blot analysis. Equal loading was confirmed by using β-Actin as a control. (**A**) Representative result of three independent experiments. (**B**) Density quantification of bands and statistical analysis. The FC cells were set as a baseline value to which FD and FR were normalized. Error bars refer to standard deviations for three independent experiments. * *p* < 0.05; *t*-test.

**Figure 3 ijms-20-01100-f003:**
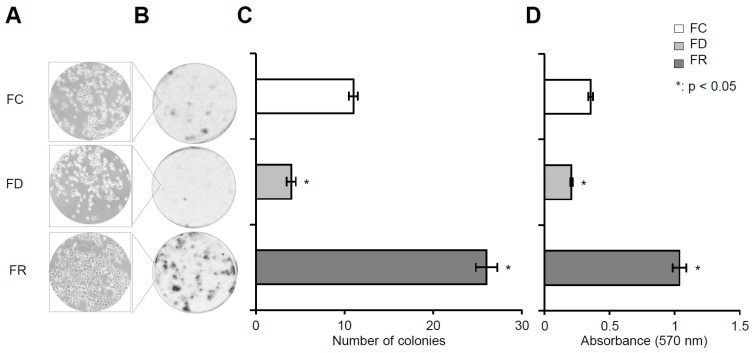
Colony formation assay. Cells were seeded and incubated in the respective folate medium. After 14 days, the cell colonies were fixed and stained with crystal violet. (**A**) Microscope images of colony-forming cells at a magnification of 10×. (**B**) Photos of crystal violet-stained cell colonies in culture wells. (**C**) Average number of colonies. (**D**) Optical density of solubilized and eluted crystal violet from stained colonies. Data shown are mean values of three independent experiments performed in duplicate. * *p* < 0.05; *t*-test HFK16E6E7 FD and FR vs. FC.

**Figure 4 ijms-20-01100-f004:**
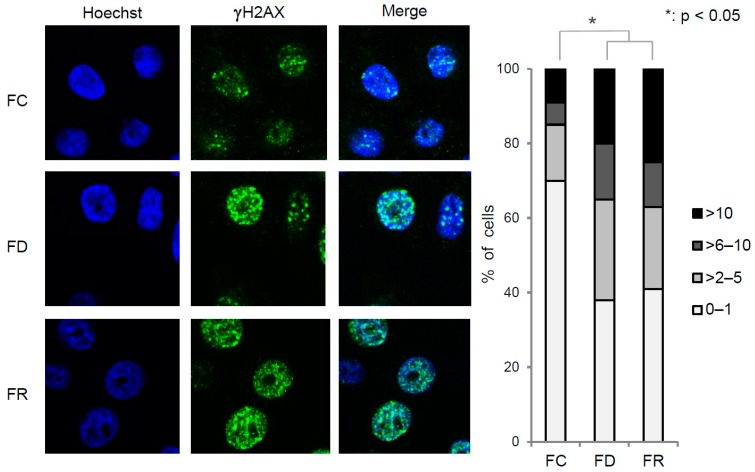
DNA double strand break detection by immunofluorescence. Immunostaining of γH2AX foci in FC, FD, and FR cells. Left panel: Representative immunofluorescence images (magnification: 60×). Cells were counterstained with Hoechst to visualize nuclei. Right panel: Percentage of cells with 0–1, 2–5, 6–10, or >10 γH2AX foci per nucleus. Histograms: Three independent experiments done in duplicates. A minimum of 80 cells per cellular condition was used for γH2AX foci counting. * *p* < 0.05; *t*-test HFK16E6E7 FD and FR vs. FC.

**Figure 5 ijms-20-01100-f005:**
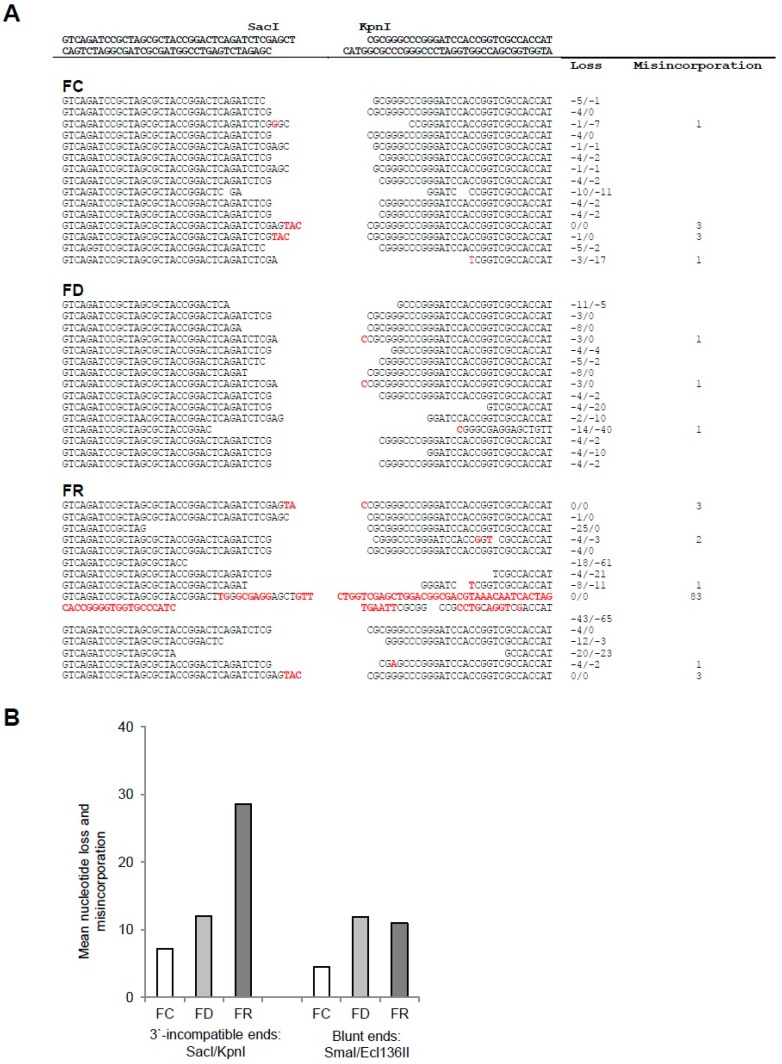
Fidelity of DNA damage repair by non-homologous end joining (NHEJ). In vivo DNA-ligation assay on FC, FD, and FR cells after transfection of linearized plasmids. (**A**) Sequences at the junction of both 3′-incompatible overhangs-end (see Appendix A for blunt-end DNA results). DNA ends of the digested plasmid are indicated in the top line. Sequences of clones for each cellular condition are indicated (empty space: Nucleotide loss; red font: Nucleotide misincorporations). The right columns indicate the total number of nucleotide losses and misincorporation for each clone. (**B**) Histograms show the nucleotide loss and misincorporation by plasmid end types and cell culture conditions. These are results from two independent experiments with a minimum of seven sequenced clones.

**Figure 6 ijms-20-01100-f006:**
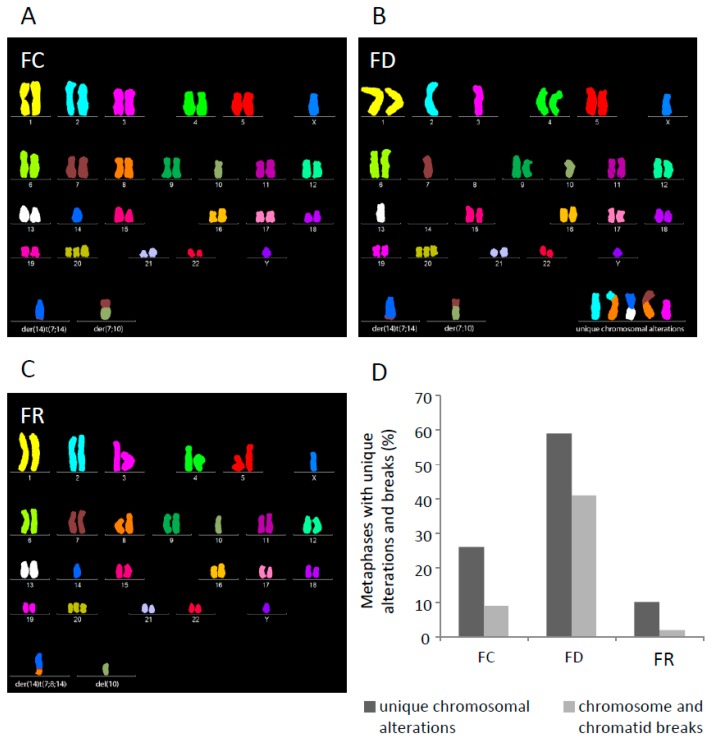
Cytogenetic analyses of FC, FD, and FR cells. (**A**) Mainline M-FISH karyotype of FC cells: 47,XY,−10,t(7;10),−14,t(7;14),+20. Derivative chromosomes der(14)t(7;14) and der(7;10) are shown in the bottom lane. (**B**) Example of M-FISH karyotype from FD cells with five non-clonal chromosomal alterations: 45,XY,−2,del(2),−3,del(3),−7,t(7;8),−8,−8,t(2;8),−10,t(7,10),−13,t(13;14),−14,−14,t(7;14),+20. Derivative chromosomes are shown in the bottom lane: der(14)t(7;14) and der(7;10) that presented in all cells (left) and unique chromosomal alterations (right). (**C**) Mainline M-FISH karyotype of FR cells: 47,XY,−10,del(10),−14,t(7;8;14),+20. Derivative chromosomes der(14)t(7;8;14) and del(10) are depicted in the bottom lane. (**D**) Rates of metaphases with unique chromosomal abnormalities and unrepaired breaks in FC, FD, and FR cells (the number of investigated metaphases, *n* = 46, was arbitrarily set as 100).

**Figure 7 ijms-20-01100-f007:**
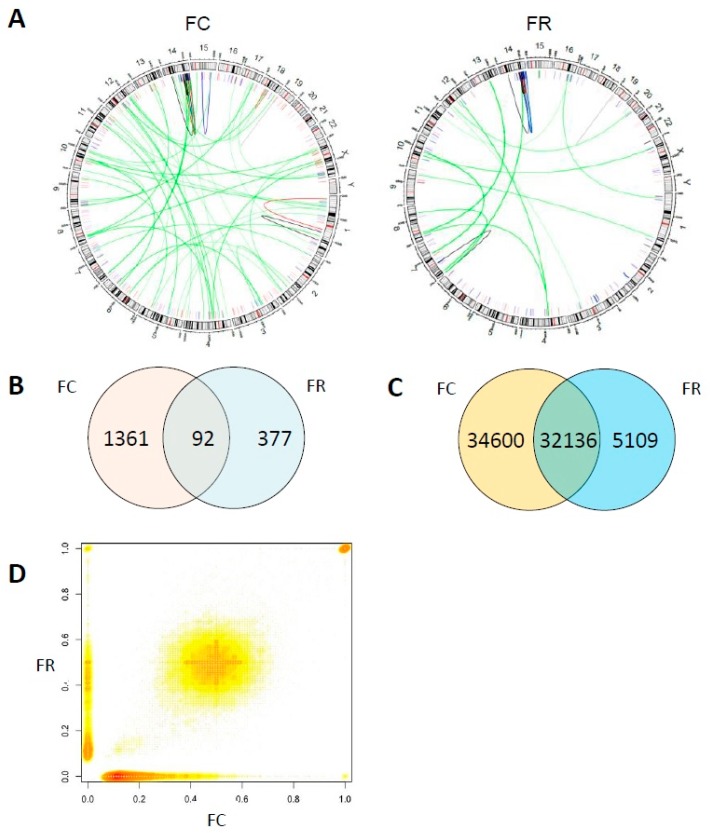
Visualization of whole genome sequence data. (**A**) Comparison between FC cells and FR subclone harboring derivative chromosome der(14)t(7;8;14). Gene duplications (red), deletions (blue), inversions (black), and translocations (green) are indicated. (**B**) Venn diagram of structural variations (SVs) detected in FC or FR sequencing results. (**C**) Venn diagram of single nucleotide variants (SNVs) detected in FC or FR sequencing results. (**D**) Scatter plot with smoothing: Comparison of minor allele frequency (MAF) for all the SNVs appearing in the cell line FC or FR clone.

**Figure 8 ijms-20-01100-f008:**
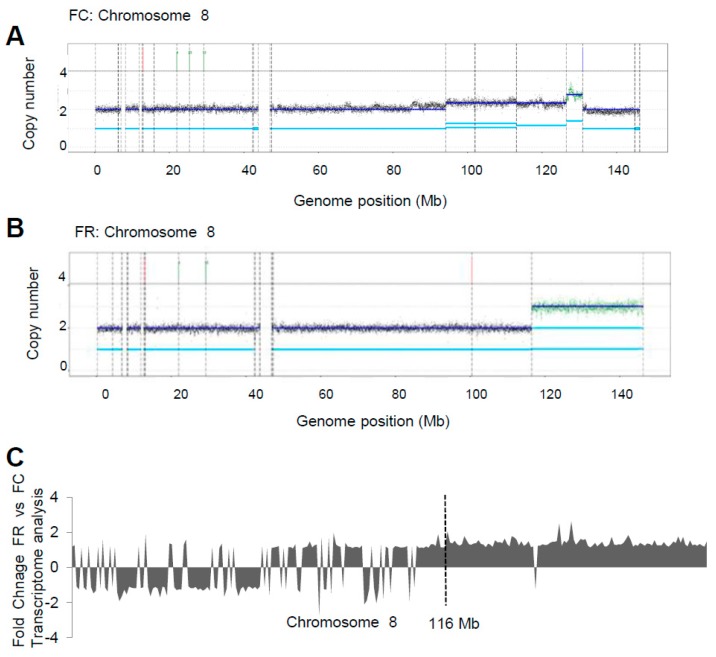
SNV allele fraction-based coverage. (**A**) and (**B**) Copy number estimation of chromosome 8 obtained using the ACEseq algorithm [30] in FC or FR cells. (**C**) Transcriptional profile of chromosome 8 genes in FR vs FC, *p* < 0.05. The ordinate expresses the fold changes of genes on chromosome 8.

**Figure 9 ijms-20-01100-f009:**
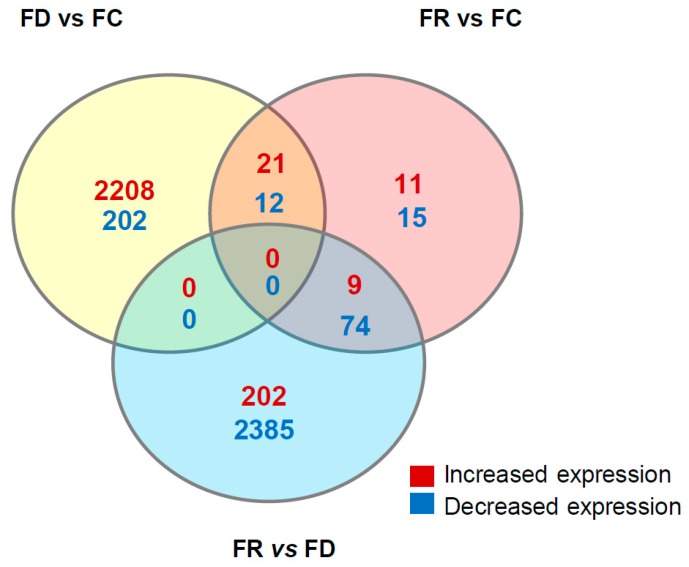
RNA-Seq comparative analysis. Venn diagram of genes significantly decreased (blue color) or increased (red color) expression in FD vs. FC, FR vs. FC, and FR vs. FD.

**Figure 10 ijms-20-01100-f010:**
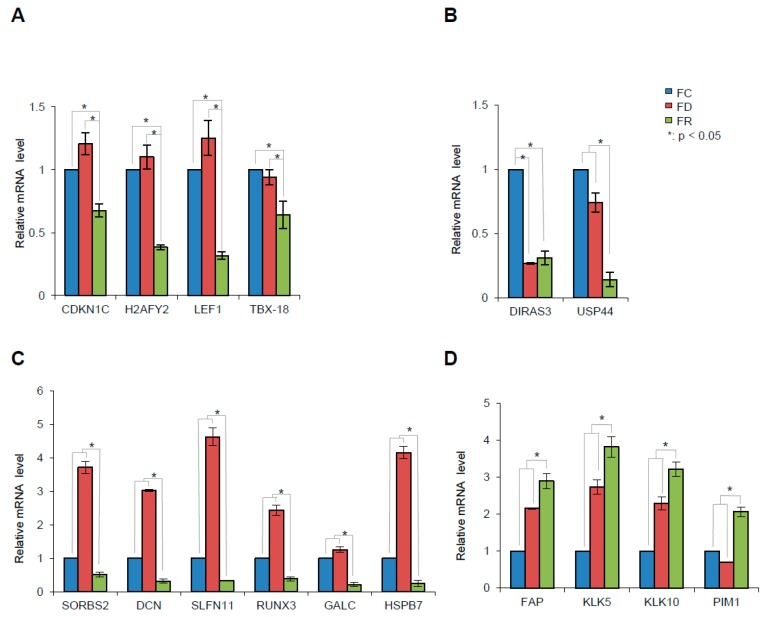
Validation of RNA-Seq selected genes by qPCR. (**A**) Genes with decreased transcription exclusively after folate repletion. (**B**) Genes with decreased transcription, both under folate deficiency and repletion. (**C**) Genes with enhanced transcription only after folate depletion. (**D**) Genes with increased transcription after folate repletion. mRNA levels were normalized to Glycerinaldehyd-3-phosphat-Dehydrogenase (GAPDH) and β-actin (ACTB) housekeeping genes expression levels. Cells grown in FC were set as 1 as reference. These results are mean values ± S.D. of three independent experiments performed in triplicate. * *p* < 0.05; *t*-test HFK16E6E7 FD vs. FC, FR vs. FC, and FR vs. FD.

**Table 1 ijms-20-01100-t001:** List of RNA-Seq genes selected for qPCR validation. Positive values: Increased expression; negative values: Decreased expression; N.S.: No significant change. The potential function of the genes is outlined with the respective references.

		Fold Change			
		RNA-Seq			
Gene Name		FD vs. FC	FR vs. FC	FR vs. FD	Comments
*SORBS2*	Sorbin And SH3 Domain Containing 2	3.5	–2.1	–7.6	Putative tumor suppressor gene involved in cervical carcinogenesis, PMID: 21602178.
*DCN*	Decorin	2.6	–2.8	–8.2	Growth suppressor of various tumor cell lines. Reduced expression or a total disappearance of DCN takes place during tumor progression, PMID: 2669749.
*SLFN11*	Schlafen Family Member 11	3.3	–4	–15.1	Putative DNA/RNA helicase, can sensitize cancer cells to DNA-damaging agents, PMID: 22927417.
*GALC*	Galactosylceramidase	N.S.	–3.4	–3.4	Lysosomal enzyme responsible for glycosphingolipids degradation by products which are important for synthesis of apoptosis. Mediator ceramide. DNA promoter hypermethylation contributes to GALC down-regulation in lung, head, and neck cancers, PMID: 26617822.
*HSPB7*	Heat Shock 27kDa Protein Family, Member 7	–2.9	–2	–9.8	Tumor suppressor gene involved in p53 signaling and down-regulated in renal cell carcinoma by hypermethylation, PMID: 24585183.
*DIRAS3*	Family, GTP-Binding RAS-Like3	–4.7	–4.6	N.S.	Tumor suppressor gene whose expression is lost in ovarian and breast cancers, PMID 16757345.
*RUNX3*	Runt-Related Transcription Factor 3	1.9	–2.6	–5.4	RUNX3 is frequently inactivated by dual mechanisms of protein mislocalization and promoter hypermethylation in breast cancer, PMID: 16818622. Association between methylation of the RUNX3 promoter and gastric cancer was reported, confirming the role of RUNX3 as a tumor suppressor gene, PMID: 21867527. RUNX3 does play important functions in immunity and inflammation and may thereby indirectly influence epithelial tumor development, PMID: 25641675.
*USP44*	Ubiquitin Specific Peptidase 44	–1.6	–3.1	–2.9	Tumor suppressor, which is epigenetically inactivated in colorectal neoplasia, PMID: 24837038. It protects against chromosome missegregation and it regulates centrosome positioning to prevent aneuploidy and suppress tumorigenesis, PMID: 23187131, PMID: 23187126.
*CDKN1C*	Cyclin-dependent kinase inhibitor 1C	N.S.	–2.1	–2.1	Negative regulator of cell proliferation. It regulates several hallmarks of cancer, including apoptosis, cell invasion and metastasis, tumor differentiation and angiogenesis, PMID: 21447370.
*H2AFY2*	H2A Histone Family, Member Y2	N.S.	–3.9	–4.2	Tumor suppressor function in lung cancer and melanoma, PMID: 26658220.
*LEF1*	Lymphoid Enhancer–Binding Factor 1	N.S.	–3.7	–4.4	Implication with AKT pathway in repairing DSB, PMID: 20956948.
*TBX18*	T-Box 18	N.S.	–2	–2.2	Tumor suppressive properties, hypermethylated in colon cancer. TBX18 inhibits growth and clonogenic survival of colon cancer cells in vitro, PMID: 20454457.
*FAP*	Fibroblast Activation Protein, Alpha	1.6	2.1	1.2	Over-expression of FAP increases tumor growth in xenografts of ovarian cancer cells, PMID: 24028972. Involved in the control of human breast cancer cell line growth and motility via the FAK pathway, PMID: 24885257.
*KLK5*	Kallikrein-Related Peptidase 5	1.7	2.5	1.3	Kallikreins are a subgroup of serine proteases having diverse physiological functions. Growing evidence suggests that many kallikreins are implicated in carcinogenesis and some have potential as novel cancer and other disease biomarkers, PMID: 19921697.
*KLK10*	Kallikrein-Related Peptidase 10	1.9	2.2	1.2	
*PIM1*	Pim-1 Proto-Oncogene, Serine/Threonine Kinase	1.8	1.2	2.2	PIM genes encode serine/threonine kinases that have been shown to counteract the increased sensitivity to apoptosis induction that is associated with MYC-driven tumorigenesis, PMID: 21150935. The PIM1 kinase promotes prostate cancer cell migration and adhesion via multiple signaling pathways, PMID: 26934497.

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
