# Peer review of "Folate Repletion after Deficiency Induces Irreversible Genomic and Transcriptional Changes in Human Papillomavirus Type 16 (HPV16)-Immortalized Human Keratinocytes"

_ijms, 2019, doi:10.3390/ijms20051100_

Reviewer 1 Report

The authors tested if folate starvation can contribute to tumorigenesis, DNA damage and chromosomal rearrangements. The idea is great but I have a concern about the experimental design. To address the importance of folate in cell biology, folate should be the only component different between FC/FR and FD groups. However, methods and materials section 4.1 clearly indicates that FBS concentrations are different in FC/FR and FD groups. FBS contains hundreds of different nutrients essential for maintaining cell stability and viability. In addition, 4.1 also lists that FC/FR contain other components that are missed in the FD medium such as insulin, EGF, cholera toxin, adenine and hydrocortisone. It is clear that there are more than just folate different between folate control and folate deficiency groups. Such a design makes it hard to have any conclusions from the experiments. 

Minor comments:

Line 68: “….may potentially drive anomalous cells towards transformation” An overstatement based on your data since you used cells already transformed. 

Figure 1B, FR curve is invisible between 0 and 15 weeks. 

Figure 2 “both oncoproteins were homogeneously expressed”

It is noticeable that E6 expression is higher in FR than FC and possibly also FD. A better way of quantification would be necessary to make such a statement. 

Figure 3 how many replicates were performed? What is the N number for the statistical analysis. Figure 3B is clearly out of focus. 

Results 2.2 

“the proportion of gH2AX foci remained even higher in FR cells” 

This is a confusing statement, does it remain or does it increase to a higher level? What does “remained even higher” mean?

From Figure 4, there is no discernable difference between FR and FD in H2AX levels. 

Figure 5

A statistical analysis should be performed. Difference between FD and FC is not prominent and may not be real unless biological replicates are done. 

Overuse of abbreviations makes reading the ms challenging and tedious. 

Author Response

Reviewer #1:

“The authors tested if folate starvation can contribute to tumorigenesis, DNA damage and chromosomal rearrangements. The idea is great but I have a concern about the experimental design. To address the importance of folate in cell biology, folate should be the only component different between FC/FR and FD groups. However, methods and materials section 4.1 clearly indicates that FBS concentrations are different in FC/FR and FD groups. FBS contains hundreds of different nutrients essential for maintaining cell stability and viability. In addition, 4.1 also lists that FC/FR contain other components that are missed in the FD medium such as insulin, EGF, cholera toxin, adenine and hydrocortisone. It is clear that there are more than just folate different between folate control and folate deficiency groups. Such a design makes it hard to have any conclusions from the experiments”. 

·We appreciate the comment of this reviewer, recognizing the significance of the media composition that may not be clear for the reader. In fact, control and depleted media have the same composition except for folate level, so we trust that the molecular effects examined in this study can be attributed to the omission of folate in “FD” cells. Nevertheless, thanks to the attention of this referee, we realized that there was indeed (unfortunately) a typo for FBS percentage in FC and FR medium, as this was 2.5% instead of 5%.

·   For clarity, we have modified this section in rephrasing again that the “FD” medium has the same composition for FBS, insulin, EGF, cholera toxin, adenine and hydrocortisone (see lines 439-443).

Minor comments:

“Line 68: “….may potentially drive anomalous cells towards transformation” An overstatement based on your data since you used cells already transformed.” 

· The referee is right with her/his concern. The word “transformation” is always a problem in terms of its vague definition. The same may be true for “anomalous” cells. Carcinogenesis is indeed a multi-step process and one has to use the right terms to describe the different progression states. However, inspecting the literature, particularly considering a review by Hanahan and Weinberg, 2011, for instance, there are many “transformation” parameters or “hallmarks” of cancer, including enhanced proliferation (Fig. 1C), enhanced plating efficiency (Fig. 3), impairment of DNA repair fidelity (Fig. 5) and reduced expression of putative tumor suppressor genes (Table 1). Since all these properties can be noticed in HPV16 E6/E7 immortalized keratinocytes (outlined in line 66) after folate repletion, we think it is appropriate to state that these cells are committed towards transformation.

· Nevertheless, we changed the original sentence “We show that supplementation to initial levels cannot compensate preceding folate deficient effects but rather force them, leading to an increased cellular proliferation, clonogenicity and selection of unique chromosomal aberrations that may potentially drive anomalous cells towards transformation” to “We show that supplementation to initial levels cannot compensate preceding folate deficient effects. Considering that cancer is a multi-step process, increased cellular proliferation, impaired DNA repair fidelity, clonogenicity and selection of unique chromosomal aberrations may potentially drive immortalized cells towards transformation” (see lines 69-73 in the revised manuscript).

“Figure 1B, FR curve is invisible between 0 and 15 weeks”. 

· A new figure is provided, now in color to make it more understandable that “FR” is deriving from “FD” as a consequence of folate repletion after deficiency.

“Figure 2 “both oncoproteins were homogeneously expressed”

It is noticeable that E6 expression is higher in FR than FC and possibly also FD. A better way of quantification would be necessary to make such a statement.” 

· Thanks for this remark. We have quantified the Western blots from 3 independent preparations. The diagrams are provided as additional panels in Fig. 2B. According to these data, there are only slight changes of E6, E7 and no changes in their major targets p53 and pRB. Hence, the visualized minor differences may be attributed to experimental variations.

·See also the new legend for Fig. 2B, where we state: “Density quantification of bands and statistical analysis. The FC cells were set as a baseline value to which FD and FR were normalized. Error bars refer to standard deviations for three independent experiments. *, p < 0.05; t-test.” (please see lines 113-117, 499)

·Nevertheless, we deleted the original sentence, now stating: “As shown in Fig. 2, there are only minor variations in E6 and E7 oncoprotein expression. p53 and pRb as major downstream targets for proteasomal degradation remained unchanged in FC, FD, and FR cells, therefore not accounting for different cellular growth behaviours (please see lines 105-107 of the revised manuscript).

“Figure 3 how many replicates were performed? What is the N number for the statistical analysis.”

·Maybe this has been overseen, but it is mentioned in the legend of Fig. 3, stating: “Data shown are mean values of three independent experiments performed in duplicate. *, p < 0.05; t-test HFK16E6E7 FD and FR vs FC.”

“Figure 3B is clearly out of focus.”

· Panel A and B only provide a representative overview of dishes. While panel A shows microscope images of colony-forming cells (at a magnification of 10x), panel B represents colonies in culture wells after staining with crystal violet (these images were obtained by scanning the plates). Nevertheless, we changed the figure legend accordingly. (see lines 123-125).

“Results 2.2  “the proportion of gH2AX foci remained even higher in FR cells” 

This is a confusing statement, does it remain or does it increase to a higher level? What does “remained even higher” mean?

· The referee is right, this sentence might be confusing. We modified it accordingly: “Formation of γH2AX foci in FR cells remained at a similar proportion as in FD cells, but is significantly higher than in FC cells. This indicates that despite the nucleotide precursor pool was restored upon folate re-supplementation, irreversible DNA damage remained.” (see lines 136-138).

“From Figure 4, there is no discernable difference between FR and FD in H2AX levels.” 

·The immunofluorescence images in Fig. 4 show the quality of γH2AX foci staining. However, after quantifying of a minimum of 80 cells (right panel), there are more γH2AX foci in FR than in FD cells. However, and this is the major outcome, a FC situation could not be reconstituted after folate repletion.

“Figure 5 A statistical analysis should be performed. Difference between FD and FC is not prominent and may not be real unless biological replicates are done.” 

·As already mentioned in the legend of Fig. 4B, the presented data are the results from two independent experiments where at least seven (for each restriction enzyme digest, in total 14) recovered plasmids were sequenced. As the focus was to measure the ability to repair sticky and blunt-end DNA, the number of nucleotides loss/misincorporation can be high while the result stays relevant. We strictly follow the protocol of Buck et al. and we have inserted again the corresponding reference (line 148).

“Overuse of abbreviations makes reading the ms challenging and tedious.” 

·We assume that the referee refers to FC, FD and FR abbreviation to which we would not see better options to avoid reiterations. Nevertheless, we have included the meaning of FC, FD and FR additionally into the abbreviation list at the end of the revised manuscript (please see line 587).

In conclusion, we thank the reviewers for their critical and helpful comments. We trust that we addressed all concerns and changed our manuscript accordingly to their suggestions

Sincerely yours,

Frank Rösl

Reviewer 2 Report

Herein is shown that folate-deficiency causes DNA damages and impairs DNA repair fidelity in HPV16-immortalized human keratinocytes, hence leading to chromosomal rearrangements. Subsequent folate repletion triggers the proliferation of keratinocytes displaying alterations and abnormal expression of either tumor suppressor or oncogenic genes. Altogether, these findings confirm a role for folate in the development and progression of HPV-associated tumors. The results from this original study, which expands previous work by others [Yang J, et al. Am J Clin Nutr 2018; Bach M,et al. Int J Mol Sci 2017; Bai LX, et al. Asian Pac J Cancer Prev 2014; Flatley JE, et al. BMC Cancer 2014; Pathak S, et al. Mol Cell Biochem 2012; Moody M, et al. Cancer Cell Int 2012; Arthur AE, et al. Nutr Cancer 2011; Mostowska A, et al. Clin Biochem. 2011 ], supply relevant information to the biomedical community. 

Minor point. Figure 2 should include the densitometric (quantitative) analysis of E6, E7, p53 or Rb protein levels (normalised to β actin) in FC, FD or FR keratinocytes. Bars should represent the mean (+ s.d.) from the three independent experiments.

Author Response

Referee #2

“Herein is shown that folate-deficiency causes DNA damages and impairs DNA repair fidelity in HPV16-immortalized human keratinocytes, hence leading to chromosomal rearrangements. Subsequent folate repletion triggers the proliferation of keratinocytes displaying alterations and abnormal expression of either tumor suppressor or oncogenic genes. Altogether, these findings confirm a role for folate in the development and progression of HPV-associated tumors. The results from this original study, which expands previous work by others [Yang J, et al. Am J Clin Nutr 2018; Bach M,et al. Int J Mol Sci 2017; Bai LX, et al. Asian Pac J Cancer Prev 2014; Flatley JE, et al. BMC Cancer 2014; Pathak S, et al. Mol Cell Biochem 2012; Moody M, et al. Cancer Cell Int 2012; Arthur AE, et al. Nutr Cancer 2011; Mostowska A, et al. Clin Biochem. 2011 ], supply relevant information to the biomedical community. 

We thank this referee for the      estimation of our work.

Minor point. Figure 2 should include the densitometric (quantitative) analysis of E6, E7, p53 or Rb protein levels (normalised to β actin) in FC, FD or FR keratinocytes. Bars should represent the mean (+ s.d.) from the three independent experiments.

Please see the comment to referee #1 that is reiterated here:

· Thanks for the suggestion. We have quantified the Western blots from 3 independent preparations. The diagrams are provided as additional panels in Fig. 2B. According to these data, there are only slight changes of E6, E7 and no changes in their major targets p53 and pRB. Hence, these differences may be attributed to experimental variations.

·See also legend for Fig. 2B, where we state: “Density quantification of bands and statistical analysis. The FC cells were set as a baseline value to which FD and FR were normalized. Error bars refer to standard deviations for three independent experiments. *, p < 0.05; t-test.” (please see lines 113-117)

In conclusion, we thank the reviewers for their critical and helpful comments. We trust that we addressed all concerns and changed our manuscript accordingly to their suggestions.

Sincerely yours,

Frank Rösl

Round  2

Reviewer 1 Report

I appreciate the authors corrected the descriptions in the MM to clarify the contents in the medium used. But it seems to me that FC and FR are maintained in medium containing 5% FBS and FD is cultured in 2.5% FBS. Am I correct? Why aren't the cells maintained in medium with the same concentration of FBS? Would 2.5% FBS difference (FBS contains tons of nutrients) contribute to the final results in addition to the difference in folate? 

Author Response

Reviewer #1:

“I appreciate the authors corrected the descriptions in the MM to clarify the contents in the medium used. But it seems to me that FC and FR are maintained in medium containing 5% FBS and FD is cultured in 2.5% FBS. Am I correct? Why aren't the cells maintained in medium with the same concentration of FBS? Would 2.5% FBS difference (FBS contains tons of nutrients) contribute to the final results in addition to the difference in folate? â€ť. 

We appreciate also the comment of this reviewer again and sorry for inconvenience.

As already outlined in our previous letter, stating: “recognizing the significance of the media composition that may not be clear for the reader. In fact, control and depleted media have the same composition except for folate level, so we trust that the molecular effects examined in this study can be attributed to the omission of folate in “FD” cells. Nevertheless, thanks to the attention of this referee, we realized that there was indeed (unfortunately) a typo for FBS percentage in FC and FR medium, as this was 2.5% instead of 5%.”

This typo has apparently been overseen after conversion of our original word file into the latest layout of the “International Journal of Molecular Sciences.

We have now corrected the M&M section accordingly (see line 432).

 In conclusion, we thank this reviewer again for her/his critical comment and trust that the manuscript should now be accepted for publication.

Sincerely yours,

Frank Rösl

Round  3

Reviewer 1 Report

n/a